# Experimental evidence of mosaic structure in strongly supercooled molecular liquids

F. Caporaletti [1,7 ✉], S. Capaccioli [2,3], S. Valenti[4], M. Mikolasek[5], A. I. Chumakov [5,6] & G. Monaco [1,8 ✉]

When a liquid is cooled to produce a glass its dynamics, dominated by the structural relaxation, become very slow, and at the glass-transition temperature $T_g$ its characteristic relaxation time is about 100 s. At slightly elevated temperatures (~1.2 $T_g$) however, a second process known as the Johari-Goldstein relaxation, $\beta_{JG}$, decouples from the structural one and remains much faster than it down to $T_g$. While it is known that the $\beta_{JG}$-process is strongly coupled to the structural relaxation, its dedicated role in the glass-transition remains under debate. Here we use an experimental technique that permits us to investigate the spatial and temporal properties of the $\beta_{JG}$ relaxation, and give evidence that the molecules participating in it are highly mobile and spatially connected in a system-spanning, percolating cluster. This correlation of structural and dynamical properties provides strong experimental support for a picture, drawn from theoretical studies, of an intermittent mosaic structure in the deeply supercooled liquid phase.

[1] Dipartimento di Fisica, Università di Trento, Povo (Trento), Italy. [2] Dipartimento di Fisica, Università di Pisa, Pisa, Italy. [3] CISUP, Centro per l'Integrazione della Strumentazione dell'Università di Pisa, Pisa, Italy. [4] Department of Physics, Universitat Politécnica de Catalunya, Barcelona, Spain. [5] ESRF-The European Synchrotron, CS40220, Grenoble Cedex 9, France. [6] National Research Center 'Kurchatov Institute', Moscow, Russia. [7] Present address: Van der Waals-Zeeman Institute, Institute of Physics/Van 't Hoff Institute for Molecular Sciences, University of Amsterdam, Amsterdam, the Netherlands. [8] Present address: Dipartimento di Fisica ed Astronomia, Università di Padova, Padova, Italy. ✉email: f.caporaletti@uva.nl; giulio.monaco@unipd.it

The timescale, $\tau_\alpha$, of structural rearrangements in a typical liquid is in the ps range. If the liquid is supercooled below the melting temperature, $\tau_\alpha$ gets longer the lower is the temperature, until the dynamics become so slow that we speak of a solid rather than of a liquid: a glass has been obtained. Conventionally, the glass transition temperature, $T_g$, is defined as the temperature where this characteristic time is 100 s. However, this is not the only characteristic timescale for the molecular dynamics of a strongly supercooled liquid. At temperatures somewhat higher than $T_g$, a second characteristic timescale associated with a secondary relaxation process appears. This secondary process, known as Johari-Goldstein or $\beta_{JG}$[1], can be considered a precursor of the glass-transition in two ways: time and temperature. In time: it anticipates the structural (or $\alpha$) relaxation that freezes in at $T_g$ by many decades, being typically in the ms-μs time range at $T_g$ and remaining active also in the glass[2]. In temperature: it decouples from the $\alpha$ process in the supercooled liquid phase at temperatures lower than $T_{\alpha\beta} \approx 1.2\ T_g$, thus anticipating the glass-transition by a stretch. On the role of the $\beta_{JG}$-relaxation in the glass-transition at least three generations of scientists have been debating since its discovery in the early seventies[1]. While its phenomenology has been largely explored, the concomitant effects of: (i) the lack of a firm microscopic theory of the glass-transition and (ii) the difficulty of experimental and numerical techniques to probe the spatial and temporal properties of this process in the relevant time-range and temperature-range, make this debate still unsettled.

From the glass-transition perspective, it is becoming more and more clear that the vitrification kinetics is not only driven by the structural relaxation, as traditionally believed, but other, faster processes must play a role to reduce the effectiveness of the structural relaxation to lock the atomic dynamics[3]. This is consistent with recent theoretical and numerical work pointing out that the free energy landscape relevant for the glassy state is much rougher than classically thought, with intra-state barriers associated with secondary relaxations[4]. In the context of these findings, it is clear that direct investigations of the $\beta_{JG}$-relaxation are of crucial importance for a better understanding of the liquid-to-glass transition.

Many basic properties of the $\beta_{JG}$ process are known, mostly thanks to a large body of dielectric spectroscopy (DS)[2,5,6] and nuclear magnetic resonance[7] data. It is e.g., nowadays understood that the $\beta_{JG}$-relaxation is strongly connected to the $\alpha$-process[2,5,6], and thus is sensitive to the glass-transition. This conclusion is in apparent contrast with the restricted re-orientational dynamics of the process[7] and with its mild Arrhenius $T$-dependence, usually interpreted as the signature of a hindered, simply activated process[2]. The few available neutron and X-ray scattering experiments also describe it, above $T_g$, as an essentially restricted if not localized molecular motion occurring mainly within the transient cages formed by the nearest-neighbors[8–12]. Sub-$T_g$ experiments[13] and atomistic simulations[14,15] support instead the idea that the $\beta_{JG}$-relaxation is rather cooperative and characterized by string-like excitations occurring via hopping between adjacent atomic sites.

Several phenomenological models of the $\beta_{JG}$ process are also available, based on an intuition of the main physical mechanism giving rise to it. For instance, according to the coupling model, the $\beta_{JG}$-relaxation consists of a distribution of elementary processes evolving with time by increasing the number of participating molecular units and culminating in the cooperative $\alpha$-relaxation: the $\alpha$-relaxation and $\beta_{JG}$-relaxation thus share many relevant properties[2]. In another model based on the hypothesis of strong dynamical heterogeneity of the supercooled liquid, the $\beta_{JG}$-relaxation originates from the population exchange between regions of tightly confined molecules and of loosely confined ones[11]. The random first-order transition theory identifies instead the $\beta_{JG}$-relaxation with cooperative rearrangements of string-like shaped regions of molecules which become more prominent with increasing temperature[16]. While these ideas are not necessarily alternative to each other, unambiguous experimental input is required as a guide among these rather different views. Combining new X-ray scattering data on 1-propanol with data available in the literature[8–10,12] and here re-analyzed, evidence shall now be provided of two striking features related to the molecules participating in the $\beta_{JG}$-relaxation: (i) their mean-squared displacement satisfies the Lindemann criterion for structural instability and (ii) their number matches the threshold for site

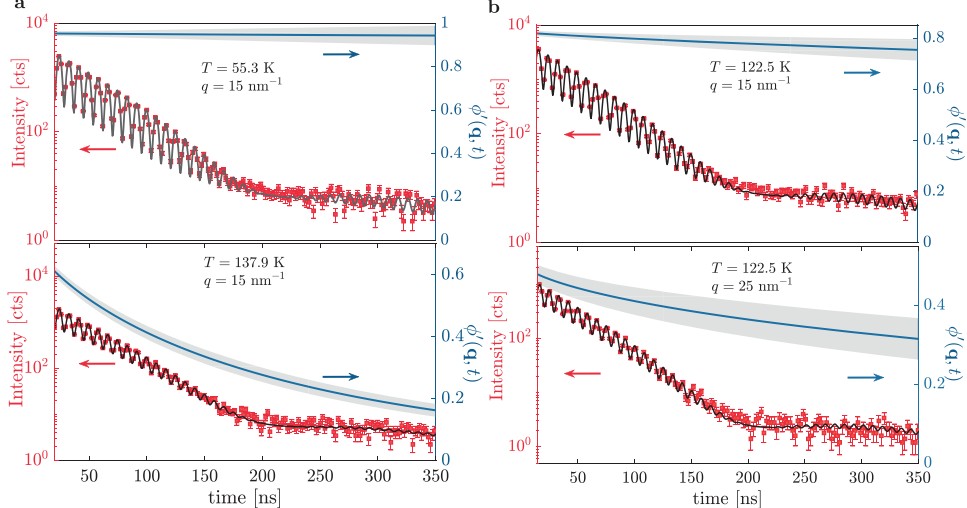

**Fig. 1 Temperature and scattering vector dependence of time-domain interferograms.** Time-domain interferometry data as a function of time (squares with ±1 SD errorbars) at different temperatures and at a fixed exchanged wave-vector ($q = 15\ \mathrm{nm}^{-1}$, roughly corresponding to the inter-molecular distance) **a** and at the same temperature $T = 122.5$ K in the supercooled liquid state and at different $q$ values **b**. The raw data have been averaged over a time range ±0.7 and ±0.9 ns, respectively, depending on the collected statistics, in order to improve the figure readability. The black solid lines are the model curves obtained from the fitting procedure. The blue solid lines are the calculated contrast functions, $\phi'$, along with the 68% confidence interval (gray area). $\phi'$ is proportional to the density correlation function.

percolation. They, therefore, form a mobile percolating, infinite cluster pervading the whole sample. This cluster evolves on the timescale of the $\beta_{JG}$-process, and careful experimental design is required to study it.

## Results

Nuclear $\gamma$-resonance time-domain-interferometry (TDI) is one of the few techniques able to probe directly microscopic density fluctuations at the timescale where the $\beta_{JG}$-process decouples from the structural one. TDI is based on an interferometric analysis of the X-rays scattered by the sample and, as such, provides information that is both time and $q$ (and therefore space) resolved ($q$ being the momentum exchanged in the scattering process)[8,12]. TDI has been here employed to study 1-propanol, a model glass-former with a genuine $\beta_{JG}$-process[17] rather coupled to the $\alpha$-relaxation. Some examples of the measured TDI beating patterns are reported in Fig. 1 (red squares) along with the fitting curves obtained using a model that accounts for one relaxation process (black lines, see "Methods" section). The beating pattern contrast function $\phi'(q,t)$ (blue solid line in Fig. 1) is proportional to the density correlation function and allows extracting its main relaxation parameters, in particular the relaxation time, $\tau$, at different $q$ values. TDI is sensitive to the fastest relaxation process active in the explored time window: given its dynamical range of fewer than two decades in time, it was not possible to detect more than one relaxation process per experimental curve even when both the $\alpha$ and the $\beta_{JG}$ process were expected to be active.

Of particular interest is the $T$-dependence of $\tau$, which was investigated at two $q$-values, 15 and 25 nm$^{-1}$, corresponding to the inter-molecular and to an intra-molecular length-scale, respectively. The measured $\tau$ values are shown in Fig. 2 along with the ones obtained by DS and relative to the reorientational dynamics (see "Methods" section and Supplementary Note 1).

The DS data provide a precise estimation of the $T$-dependence of the $\alpha$-process and $\beta_{JG}$-process characteristic times (gray and green dash-dotted lines in Fig. 2). These $T$-dependencies were then scaled onto the TDI data. This procedure, along with the study of the $q$-dependence of the relaxation time (see below), allowed us to unambiguously associate the process appearing in the density fluctuations to the $\alpha$-relaxation or $\beta_{JG}$-relaxation, see also Supplementary Note 2 for additional support to this interpretation.

A clear change in the $T$-dependence of $\tau$ measured by TDI occurs around $T_{\alpha\beta} \simeq 131$ K at $q = 25$ nm$^{-1}$. At $q = 15$ nm$^{-1}$ the crossover from the $\alpha$ to the $\beta_{JG}$-process is instead too weak to be appreciated. Interestingly, at $q = 15$ nm$^{-1}$ the $T$-dependence of $\tau$ can also be described accounting only for that of the $\beta_{JG}$-relaxation (blue dashed line in Fig. 2): a similar match cannot be obtained instead considering only the $\alpha$-process. It is then evident that, at least for $T \leq 131.4$ K, TDI is sensitive to the $\beta_{JG}$-process both at the inter and intra-molecular lengthscales. This provides broader validity to similar observations for another monoalcohol[12].

Further insights can be gained from the $q$-dependence of the relaxation parameters which was investigated at two temperatures: 131.4 and 122.5 K (see Fig. 3a). At the former one, slightly higher than the decoupling temperature $T_{\alpha\beta}$, the $\alpha$-relaxation is expected to be dominant, whereas at the latter one Fig. 2 shows that TDI is sensitive only to the $\beta_{JG}$-process. In order to facilitate the comparison to the DS results, the TDI characteristic times have been converted to those, $\tau_p$, corresponding to the peak of the susceptibility function (see "Methods" section). The obtained $\tau_p$ values are plotted in Fig. 3b along with the curves obtained from fitting to them a power-law $\tau_p \propto q^{-n}$. The power-law exponents,

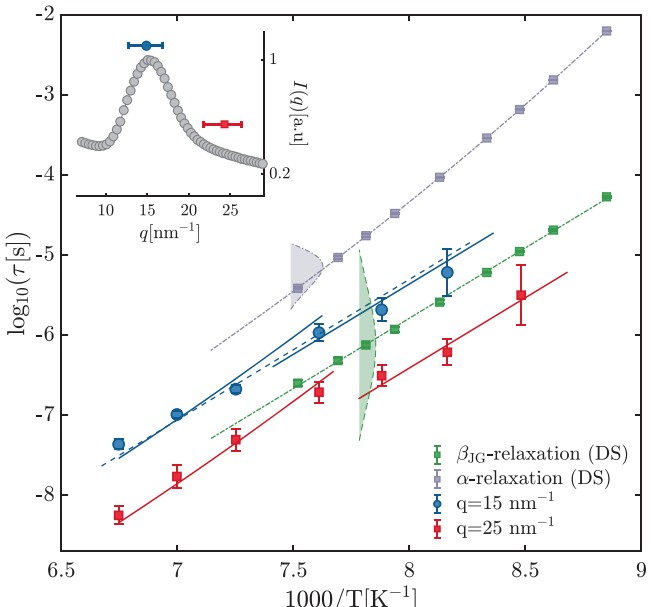

**Fig. 2 Relaxation map from dielectric spectroscopy and time-domain interferometry measurements.** Inverse temperature dependence of the relaxation time ($\tau$) measured by dielectric spectroscopy (gray and green squares with ±1 SD errorbars) and time-domain interferometry at two different $q$-values: 15 (blue circles with ±1 SD errorbars) and 25 nm$^{-1}$ (red squares with ±1 SD errorbars). The gray and green dash-dotted lines are fit to the dielectric data. The $\alpha$-relaxation is described using the Vogel-Fulcher-Tamman equation $\tau = \tau_0 \exp(D T_0/(T - T_0))$ ($D = 37(2)$, $\tau_0 = 10^{-15.4(2)}$s, $T_0 = 51(1)$ K) (gray dash-dotted line) while the $\beta_{JG}$ relaxation using the Arrhenius $T$-dependence with a reduced activation energy $E_{\beta_{JG}}/k_B = 4.04(4) \times 10^3$ K (green dash-dotted line). The blue and red solid lines are the model results fitted to the dielectric spectroscopy data and scaled to match the time-domain interferometry data. The blue dashed line is the best match of the $\beta_{JG}$ $T$-dependence to the whole set of time-domain interferometry data at 15 nm$^{-1}$. The gray and green areas delimited by dashed lines at $1000/T = 7.5$ and 7.8 K$^{-1}$ are the distributions of relaxation times, $G(\ln \tau)$, associated with the $\alpha$-process and $\beta_{JG}$-process, respectively, and extracted from dielectric spectroscopy investigations. The base widths of the two areas correspond to the FWHM of the two distributions. Inset: diffuse scattering pattern of 1-propanol at $T = 122.5$ K, with the indication of the $q$ values and of the corresponding ranges covered in the time-domain interferometry measurements reported in the main figure.

$n$, obtained from the fits, namely $n = 2.0(7)$ at $T = 131.4$ K and $n = 3.9(9)$ at $T = 122.5$ K (see Fig. 3c), strongly support that for $T > T_{\alpha\beta}$ the $\alpha$-relaxation, with a quadratic $q$-dependence of the characteristic time, dominates the microscopic dynamics, whereas below $T_{\alpha\beta}$ the characteristic super-quadratic $q$-dependence of $\tau_p$ confirms that the $\beta_{JG}$-relaxation is rather observed.

## Discussion

The $\alpha$-process in 1-propanol is still diffusive around $T_{\alpha\beta}$ while the microscopic dynamics associated with the $\beta_{JG}$-process is clearly restricted ($n > 2$) already at a relatively high $T$ (122.5 K $\simeq 1.25\ T_g$), giving broader significance to similar observations reported for the other two glass-formers studied using TDI: 5M2H[12] and OTP[8,9].

There are then some interesting common features for the three glass-formers investigated by TDI: (i) density fluctuations are coupled to the $\beta_{JG}$ relaxation at/close to both the inter and intra-molecular length-scale; (ii) the characteristic times of the $\beta_{JG}$ relaxation appearing in the density fluctuations show a strong $q$-dependence, indicative of a restricted dynamics, and (iii) cross the

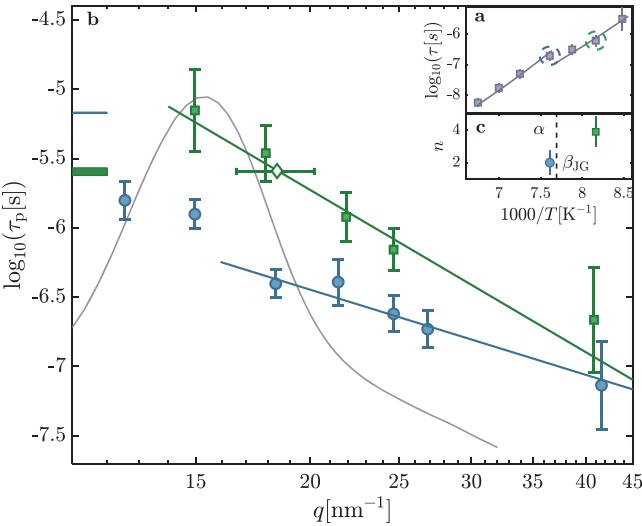

**Fig. 3 Scattering vector dependence of the relaxation time of density fluctuations. a** temperature dependence of the relaxation time at $q = 25$ nm$^{-1}$, as in Fig. 2. The green and blue circles show the two temperatures at which the $q$-dependencies in the main figure are reported. **b** wavenumber ($q$) dependence of the characteristic time ($\tau_p$) of density fluctuations identified by the peak position of the susceptibility function at $T = 131.4$ K ($\alpha$-relaxation, blue circles with ±1 SD errorbars) and 122.5 K ($\beta_{JG}$-relaxation, green squares with ±1 SD errorbars). The solid-lines are obtained fitting a power-law $\tau_p \propto q^{-n}$ to the experimental data. This simple model can account for the $q$-dependence of the $\beta_{JG}$-relaxation at $T = 122.5$ K in the whole explored $q$-range. At $T = 131.4$ K, where the $\alpha$-relaxation is the dominant process, $\tau_p$ shows an oscillation close to the peak of the $S(q)$ that can be associated with the well-known de Gennes narrowing effect[42]. A power-law is indeed able to well reproduce the experimental data only starting from $q = 18$ nm$^{-1}$. The blue and green boxes indicate the values of the corresponding relaxation times measured by dielectric spectroscopy. Their widths show the corresponding uncertainty (±1 SD). For $T = 122.5$ K the reported value of $\tau$ is the mean of our dielectric spectroscopy result and of that from ref. [36]. The open diamond with horizontal ± 1 SD errorbar is the $q$-value, $q_{DS}$, at which the TDI and DS relaxation times match at $T = 122.5$ K. The diffuse scattering pattern at 122.5 K (gray line) is rescaled and reported on the same plot for the sake of comparison. **c** the $n$ values extracted from the power-law fits are plotted as a function of the inverse temperature. The errorbars correspond to ±1 SD.

dielectric $\tau_{\beta_{JG}}^{DS}$ at a $q$-value, $q_{DS}$, somewhat larger than $q_{max}$. The length $\frac{1}{q_{DS}}$ identifies the characteristic scale for the center-of-mass motion within the $\beta_{JG}$-relaxation[12]. More in detail, it is possible to relate $q_{DS}$ to the molecular root-mean-squared displacement within an anomalous diffusion model[18] (see discussion in the Supplementary Note 3) as:

$$\Delta r_{JG} = \sqrt{\langle r^2(\tau_{\beta_{JG}}^{DS})\rangle} = \frac{\sqrt{6}}{q_{DS}} . \tag{1}$$

Here $\Delta r_{JG}$ has to be regarded as the most probable displacement of the molecules participating in the $\beta_{JG}$-relaxation at $\tau_{\beta_{JG}}^{DS}$. Figure 4a shows $\Delta r_{JG}$ for 1-propanol (red symbol), 5M2H (blue symbols), and OTP (green symbols, see Supplementary Note 4) estimated from the $q$-dependencies reported in Fig. 3 and in refs. [8,9,12] and normalized to the corresponding average inter-molecular distance (center-of-mass to center-of-mass), $r_p = v_{mol}^{\frac{1}{3}}$, where $v_{mol}$ is the molecular volume. Figure 4b reports the corresponding relaxation times of the three samples probed by DS and depolarized dynamic light-scattering as a function of $T_g/T$ and scaled by the value of $\tau_{\beta_{JG}}^{DS}$ at $T_g$. Figure 4a shows that $\Delta r_{JG}$ is

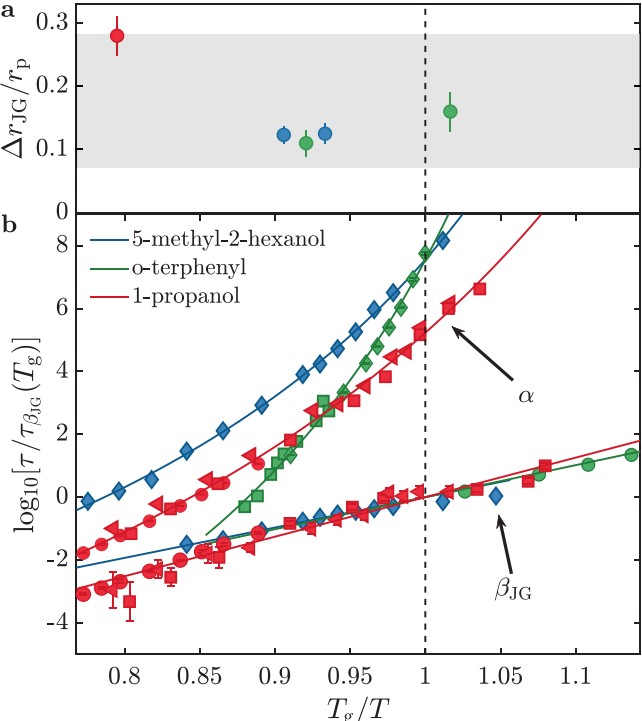

**Fig. 4 Root mean squared displacement for the molecules participating in the $\beta_{JG}$ process at $\tau_{\beta_{JG}}^{DS}$. a** Characteristic center-of-mass root-mean-squared displacement at $\tau_{\beta_{JG}}^{DS}$ for the molecules participating in the $\beta_{JG}$-relaxation rescaled to the corresponding inter-molecular distance (center-of-mass to center-of-mass). Red circles: 1-propanol; blue circles: 5M2H; green circles: OTP. The gray area shows the range of typical values for the Lindemann criterion in crystals (see ref. [20] and references therein) expressed in terms of root-mean-squared displacement. **b** $T_g$-rescaled inverse $T$-dependence of the $\alpha$-relaxation and $\beta_{JG}$-relaxation times ($\tau$) measured by dielectric spectroscopy and/or depolarized dynamic light-scattering for 5M2H (blue diamonds,[12]), 1-propanol (red left-pointing triangles and squares,[36]; red circles, this work) and OTP (green squares,[43]; green circles,[1]; green diamonds, this work). To facilitate the comparison the relaxation times have been scaled to the value of the corresponding $\tau_{\beta_{JG}}^{DS}$ at $T_g$. All errorbars shown in panels **a** and **b** correspond to ±1 SD.

larger for 1-propanol ($\simeq 28\%$) than for 5M2H and OTP ($\simeq 12\%$). This difference can be related to the higher normalized temperature at which the dynamics of 1-propanol were investigated, i.e., $T/T_g \simeq 1.25$. In all cases, $\Delta r_{JG}$ amounts to a non-negligible fraction of the average inter-molecular distance, a result evocative of the Lindemann criterion for the stability of crystalline solids[19]. In fact, all the estimated values for $\Delta r_{JG}/r_p$ fall inside the gray band in Fig. 4a which enlightens the range of typical values for the Lindemann criterion in crystals (see ref. [20] and references therein) expressed in terms of root-mean-squared displacement. This suggests that the restricted molecular motion associated with the $\beta_{JG}$-relaxation corresponds, in average, to locally unstable cages: the molecules participating in the $\beta_{JG}$-process can be seen as 'uncaged' at $\tau_{\beta_{JG}}^{DS}$ and are characterized by higher-than-average mobility as they can eventually sub-diffuse to longer distances prior the onset of the $\alpha$-relaxation.

Further insights on the structural properties of the $\beta_{JG}$-relaxation are provided by its microscopic relaxation strength, $f_q^{JG}$. This information is hard to obtain even for the dynamic range of TDI and the only available estimate is for 5M2H[12] where it was found that around $q_{DS}$ (31 nm$^{-1}$ < $q$ < 40 nm$^{-1}$) and above $T_g$ (1.07 $T_g$ < $T$ < 1.21 $T_g$) $f_q^{JG} \simeq 0.25$: one molecule out of four, on average, participates to the $\beta_{JG}$-process at $\tau_{\beta_{JG}}^{DS}$.

It is interesting to read this value in the light of the number of nearest neighbors ($z$) for the samples here considered. $z$ can be estimated from the radial pair distribution function, $g(r)$: for 1-propanol and OTP the room temperature values are $z \simeq 12$ (D.T. Bowron, private communication) and $\simeq 15$[21], respectively (see Supplementary Note 5). No $g(r)$ data are instead available in the literature for 5M2H but we can expect its number of nearest neighbors to be similar to that of 1-propanol, given the similarity of their structure. The available values of $z$ are close to that for an fcc lattice, and remarkably lattices with such connectivity present a threshold for site percolation ($p_c^{f.c.c} = 0.198$[22]) which is comparable with $f_q^{JG}$. The molecules participating in the $\beta_{JG}$-relaxation must then be spatially connected in a percolating (or close to percolation) cluster pervading the whole sample, see Fig. 5. Clearly, this spatial connection does not imply dynamical correlations among all the molecules participating in the $\beta_{JG}$-process. In fact, though the $\beta_{JG}$-relaxation might be cooperative to some extent, as our results for 5M2H[12] and 1-propanol also suggest, its strong sub-diffusivity clearly evidences that it is of prevalent local nature.

The present results establish an interesting correlation between spatial and temporal properties of the molecules participating in the $\beta_{JG}$ process: the most mobile molecules (on a time scale that precedes the $\alpha$-process) are spatially distributed as a percolating cluster. In other terms, the $\beta_{JG}$ process marks the development of a mosaic state in the undercooled phase with patches of less mobile molecules separated by an intermittent and ever-changing network of more mobile ones. These results, obtained in the

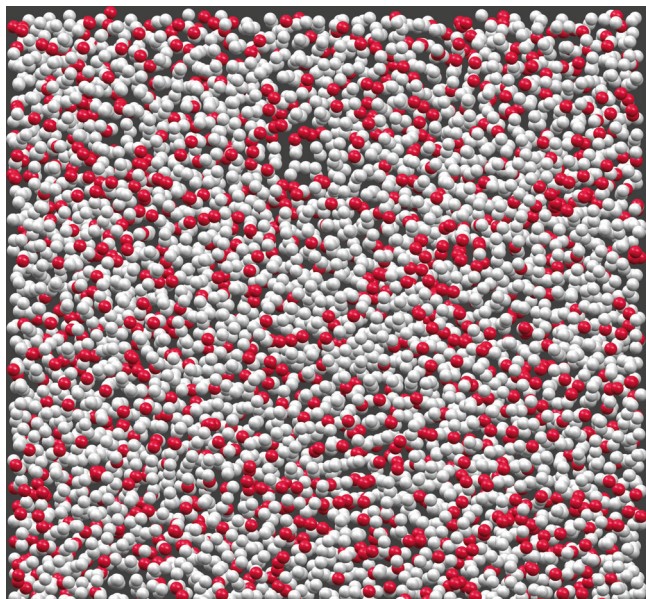

**Fig. 5 Sketch of the spatial distribution of the molecules participating in the $\beta_{JG}$-process at a given time.** The molecules undergoing the $\beta_{JG}$-relaxation (red spheres) are highly mobile and, after a time of the order of $\tau_{\beta_{JG}}^{DS}$, perform larger spatial excursions than the rest of the molecules (white spheres). These spatial excursions are, on average, of the order of those prescribed by the Lindemann criterion for crystal instability[19]. These "uncaged" molecules form a (close to) percolating cluster. The sketch is based on an experimentally measured configuration for a colloidal glass of hard spheres (volume fraction 0.61) reported in ref. [44] but has no direct connection to the discussion reported in that study. The cluster of mobile molecules was obtained by randomly selecting particles with a probability of 0.25, according to the here discussed estimate for the fraction of molecules participating in the $\beta_{JG}$-relaxation at $\tau_{\beta_{JG}}^{DS}$ (relaxation strength).

deeply supercooled liquid phase, give then a new perspective to concepts used to imagine the dynamical heterogeneities which develop on lowering the temperature towards $T_g$[23–25]: while the size of the patches of the mosaic structure shows in the investigated cases a mild temperature dependence[26–28], the development of a percolating, infinite cluster at temperatures below $T_{\alpha\beta} \approx 1.2 \ T_g$ signaled by the $\beta_{JG}$-process is a very suggestive result supporting the idea that a dynamical transition is taking place there. This point is in line with the basic idea underlying the random phase order transition theory[16,29].

## Methods

**Nuclear $\gamma$-resonance time-domain interferometry measurements.** The nuclear $\gamma$-resonance TDI experiments were performed at the nuclear resonance beamline ID18[30] of the European Synchrotron Radiation Facility (ESRF) in Grenoble (F) employing an optimized implementation of what is originally described in ref. [31].

TDI experiments are designed to detect X-ray scattering using an interferometer that allows probing density fluctuations in the 10 ns–10 μs time range[8,10,12]. Two single-line $^{57}$Fe-containing absorbers were therefore installed upstream of the sample, on the incoming beam, and downstream of the sample, on the scattered beam, to provide the probe and reference beams of a time-domain interferometer. In order to obtain different excitation energies, the probe absorber was mounted on a velocity transducer and driven at the constant velocity $v \sim 10$ mm/s with a relative accuracy better than 0.1%. The resulting shift was of $\hbar\Omega = 105 \ \Gamma_0$, where $\Gamma_0 = 4.66$ neV is the natural linewidth of the first excited state of $^{57}$Fe. The incident X-ray radiation used to excite the nuclear resonance of the absorbers was characterized by a bandwidth of 2.5 meV at the energy of the first nuclear transition of $^{57}$Fe at 14.412 keV. The photons quasi-elastically scattered by the sample at two different scattering vectors $q = 2k_0 \sin(\theta/2)$, where $\theta$ is the scattering angle and $k_0 = 73$ nm$^{-1}$ is the wave-vector of the nuclear fluorescence from the first excited state of $^{57}$Fe, were simultaneously collected by two avalanche photodiode (APD) detectors. The setup was designed to span the $q$-range 9–42 nm$^{-1}$.

The TDI interferograms were analyzed as reported in the literature[12,32–34]. More precisely, in TDI experiments with identical single-line nuclear absorbers the time evolution of the intensity emerging from the interferometer is described by[12,32,33]:

$$I(q,t) = |R(t)|^2 [1 + \cos(\Omega t)\phi'(q,t)]. \quad (2)$$

Here $R(t)$ is the time response of the nuclear absorbers[12,32,33] and $\phi'(q,t)$ (blue solid line with gray area in Fig. 1) is the contrast function[12]. $\phi'(q,t)$ is related to the density correlation function (also known as normalized intermediate scattering function), $\phi(q,t)$, via:

$$\phi'(q,t) = \phi(q,t)\frac{2}{1 + f_{\Delta E}(q,T)}, \quad (3)$$

where $f_{\Delta_E}$ depends both on the experimental set-up and on sample properties[32,34]. $\phi'$ has been modeled using the KWW function[2] in order to introduce a minimum bias in the fitting procedure[8,12]:

$$\phi'(q,t) = f'_q \exp\left[-\left(\frac{t}{\tau}\right)^{\beta_{KWW}}\right]. \quad (4)$$

Here $f'_q$ is the initial beating pattern contrast (see Supplementary Note 6 for the obtained results), $\tau$ is the relaxation time of density fluctuations and $\beta_{KWW}$ is the stretching parameter. In the fitting procedure $\beta_{KWW}$ was fixed to an average value obtained from DS measurements ($\langle\beta_{KWW}\rangle = 0.66$) for $q > q_{max}$, similarly to what reported in refs. [8–10,12]. Concerning the data collected at $q_{max}$, $\beta_{KWW}$ has been increased by 20% ($\langle\beta_{KWW}\rangle = 0.79$) in order to take into account its $q$-dependence, consistently to what discussed in ref. [12]. $\phi'$ has also been modeled using the Mittag–Leffler function[35], the Fourier transform of the Cole–Cole function well known to properly describe the $\beta_{JG}$ relaxation in DS data[2]. This approach is delicate given the fact that (i) the TDI signal covers less than two decades in time and (ii) the Fourier transform of the Cole–Cole function has a very stretched tail. In other terms, the measured TDI signal is in general not very sensitive to the parameters of the (Fourier transform of) the Cole–Cole function. However, for the data-sets where this happens to be the case, modeling the TDI signal within the $\beta_{JG}$-relaxation using the Cole–Cole susceptibility provides results very close to those obtained using the KWW function for what concerns both the quality of the fit and the estimated parameters, which turn out to be in mutual agreement within one standard deviation.

**1-propanol.** The 1-propanol sample was purchased from Sigma Aldrich (anhydrous, 99.7% pure) and used as received. 1-propanol, differently from other glass-formers investigated by TDI[8,12], is characterized, above $T_g = 97$ K, by rather coupled $\alpha$ and $\beta_{JG}$ processes[36], thus offering the possibility to study the $\beta_{JG}$-relaxation in a regime still unexplored by density fluctuations. The TDI experiment on 1-propanol has been carried out in the temperature interval from 1.44 $T_g$ down to

1.16 $T_g$ and at scattering vectors ($q$) ranging from the peak of the structure factor $q_{max} = 15\,nm^{-1}$ up to 40 nm[1]. The temperature of the sample was controlled using a He-flow cryostat with ±0.1 K stability.

**Dielectric spectroscopy measurements**. The complex permittivity of the sample was measured in the range 10 mHz–10 MHz using a lumped impedance technique and the Novocontrol Alpha-Analyzer, whereas in the range 1–3 GHz using the coaxial reflectometric technique[37] employing the Agilent 8753ES network analyzer. The dielectric cell consisted of a parallel plate capacitor with a silica spacer and filled with the sample in the liquid state. The temperature of the sample was controlled using a dry nitrogen-flow Quatro cryostat with a temperature accuracy better than 0.1 K.

The measured permittivity function $\epsilon(\nu)$ was analyzed fitting simultaneously its real $\epsilon'(\nu)$ and imaginary part $\epsilon''(\nu)$, where $\nu$ is the frequency. The function used for the fits is:

$$\epsilon(\nu) = \frac{\Delta\epsilon_D}{1 + j2\pi\nu\tau_D} + \Delta\epsilon_\alpha L_{j2\pi\nu}\left\{ -\frac{d}{dt}\exp\left[-\left(\frac{t}{\tau_\alpha}\right)^{\beta_{KWW}}\right]\right\}$$
$$+ \frac{\Delta\epsilon_{\beta_{JG}}}{1 + \left(j2\pi\nu\tau_{\beta_{JG}^{DS}}\right)^a} + \frac{\sigma}{j2\pi\nu\epsilon_0} + \epsilon_\infty, \tag{5}$$

where the first Lorentzian term accounts for the Debye relaxation; the second Kohlrausch-Williams-Watts (KWW) term for the $\alpha$-relaxation; the third one is the Cole–Cole function that describes the $\beta_{JG}$-relaxation; the fourth term accounts for the d.c. conductivity contribution; and the last one is for the induced polarization dielectric constant. $\Delta\epsilon_{D,\alpha,\beta_{JG}}$ are the dielectric relaxation strengths of the different processes. $L_{j2\pi\nu}\{\}$ is the Laplace transform evaluated at $j2\pi\nu$.

Treating the $\alpha$ and $\beta_{JG}$-processes as statistically independent is a common practice in the literature when the latter is not resolved. Nevertheless one has to be aware that the two processes are strongly connected in properties and not independent, and their cross-correlation terms could play a role. Our data are in agreement with those reported in ref. [36], as can be observed from Fig. 4 in the main text, and also with those in ref. [38]. More details on the model used here can be found in ref. [39]. An example of a dielectric spectrum measured for 1-propanol along with the fitting curve is shown in Supplementary Fig. 1.

**Comparison between TDI and DS data**. Since the TDI and the DS data were fitted employing different models, namely the KWW equation (Eq. (4)) for the TDI data and the Cole–Cole equation (third term of Eq. (5)) for the DS data, the comparison between the two timescales, required to estimate $q_{DS}$, was performed using in both cases the characteristic time identified by the position of the maximum of the corresponding susceptibility, $\chi''(\omega)$. To this aim, starting from the $\tau$ values obtained from fitting Eq. (4) to the TDI signal, the corresponding susceptibility was numerically computed in order to obtain the characteristic time $\tau_p$ defined as:

$$\tau_p = \frac{1}{\omega_p}, \tag{6}$$

where $\omega_p$ is the maximal loss angular frequency for that susceptibility. $\omega_p$ is displaced to slightly lower frequencies with respect to $1/\tau$. The susceptibility corresponding to Eq. (4) was computed using the algorithm described in ref. [40] since the KWW equation has no analytical Fourier/Laplace transform.

In the case of the Cole–Cole expression, $\tau_{\beta_{JG}}^{DS}$ is already equal to $\frac{1}{\omega_p}$.

**Mean square amplitude and mean-squared displacement**. The values indicated in ref. [20] for the Lindemann criterion, i.e., molecular displacements comprised between 5% and 20% of the average intermolecular distance, refer to the mean-square amplitude (MSA):

$$MSA = \sqrt{\langle|\mathbf{r} - \langle\mathbf{r}\rangle|^2\rangle}, \tag{7}$$

where $\mathbf{r}$ is the molecular position, $\langle\mathbf{r}\rangle$ is the average position and $\langle\cdot\rangle$ denotes the ensemble average. $\Delta r_{JG}$, as explained in the main text, is instead evaluated from the mean-squared displacement which is defined as:

$$MSD = \langle|\mathbf{r}(t) - \mathbf{r}(0)|^2\rangle. \tag{8}$$

Under the hypothesis that the molecular positions at times 0 and $t$ are uncorrelated, the MSD and the MSA are related by[41]:

$$MSA = \frac{\sqrt{MSD}}{\sqrt{2}}. \tag{9}$$

In order to perform the comparison shown in Fig. 4, the range of values reported in ref. [20] has therefore been multiplied by $\sqrt{2}$.

## Data availability

The data that support the findings of this study are available from the corresponding authors upon reasonable request. The data used to produce Fig. 5 have been taken from http://www.physics.emory.edu/faculty/weeks/data/mono.html.

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

## Acknowledgements

We acknowledge the ESRF for the provision of the synchrotron radiation facilities at the Nuclear Resonance beamline ID18, and we thank J.-P. Celse for his help in the implementation of the experimental set-up. We also thank Prof. Kia L. Ngai (CNR-IPCF, Pisa, IT) for interesting discussions on the nature of the $\beta_{JG}$ relaxation, Prof. G.P. Johari (McMaster University, Hamilton, CA) for his comments to the manuscript and Dr. D.T. Bowron (STFC, ISIS Neutron and Muon Source, UK) for sharing the intermolecular pair correlation function data of 1-propanol. S.C. and S.V. thank CISUP for the access to dielectric spectroscopy laboratory facilities. The research at the University of Pisa was funded by the project PRA-2018-34 "ANISE".

## Author contributions

The project was conceived by F.C. and G.M. The TDI experiments were performed by F.C., S.C., S.V., M.M., A.I.C., and G.M. The TDI data were analyzed by F.C. with inputs from A.I.C. and G.M. The DS experiment and the related data analysis were performed by S.C. and S.V. All the authors discussed the results. The paper was written by F.C. and G.M. with inputs from all the authors.

## Competing interests

The authors declare no competing interests.
