## [Peer Review File · Nature Communications]

REVIEWER COMMENTS

Reviewer #3 (Remarks to the Author):

The manuscript reports an in-depth characterization, with both temporal and spatial resolution, of the dynamics of molecular glass former 1-propanol by gamma-resonance time-domain-interferometry (TDI) and dielectric spectroscopy (DS). Previously published data on other glass formers are considered too. The main outcome of the study is that the Johari-Goldstein (JG) beta relaxation is not just a local relaxation whose effects are spatially limited, but it is rather associated to a percolating cluster made by highly mobile connected molecules. Though several arguments are employed to attain this conclusion, the central one is based on the observation that the most probable mean-squared displacement of molecules participating to the JG beta relaxation – taken from the maximum of the imaginary part of the dielectric susceptibility – is large enough to fall in the range where structural instability takes place according to the Lindemann criterion.

The conclusions of the present study go beyond the traditional beliefs on the often neglected role in the glass transition of relaxations beyond the main alpha relaxation. The arguments provided to reach these conclusions have solid physical bases. The manuscript can be recommended for publication. However, before that the authors must address a number of concerns and modify the manuscript accordingly.

1. The premise of the work is based on whether the (JG) beta relaxation can be considered a mere spectator or rather is a driver of the glass transition. The conclusion that the latter holds must be contextualized within our current knowledge showing that not only the alpha process determines the glass transition: *Sci. Adv.* 6, 1454 (2020); and that the landscape is much rougher than is classically assumed considering only the alpha relaxation as a driver of the glass transition: *Nat. Comm.* 5, 3725 (2014).
2. TDI experiments are conducted in proximity of the alpha-beta splitting. Why none of the density correlation functions under any condition provide any evidence for both the two relaxations?
3. A crossover from alpha to beta relaxations is claimed at 15 nm⁻¹. However, I see no unambiguous signature of this whatsoever. Here it is safe to avoid speculating on such crossover considering that later in the manuscript (Fig. 3) the alpha and beta process dominance in each temperature range becomes clear.
4. At the beginning of page 3, I would rather say that the q dependence of tau demonstrates the alpha and beta relaxation nature of the processes observed at 131.4 and 122.5 K.
5. The average intermolecular distance is obtained from the molecular volume. This connection must be discussed in details.
6. All considerations regarding the mean-squared displacement of molecules taking part to the JG beta relaxation only account for the most probable relaxation time is DS. Why not considering the mean-squared displacement of the whole distribution of tau_betaJG? This should provide additional

important information on the fraction of the beta JG relaxation fulfilling the Lindemann criterion for structural instability.

7. I found rather peculiar that the authors employ the Cole-Cole equation to fit the beta JG relaxation and then claim that the beta JG relaxation can be described by the KWW equation. It is well known that the two descriptions are not equivalent (rather the Havriliak-Negami equation, under certain conditions, is equivalent to the KWW equation). If the beta JG relaxation is symmetric, I see no way how the KWW could be valid.

8. Anyhow the all β_{KWW} values must be provided. If, as reported in the SI, $n \cdot \beta_{KWW} = 2$, then β_{KWW} should be in all cases approximately 0.5.

Reviewer #5 (Remarks to the Author):

The manuscript deals with the solution of a problem in the physics of liquids and glasses, namely the microscopic basis for the so-called Johari-Goldstein relaxation, also known as beta-relaxation. This characterizes the dynamical behavior of a liquid upon the transition into the glassy state. This dynamics sets in significantly before a glass turns into a liquid when being warmed up, i.e., the beta relaxation precedes the so-called alpha-relaxation where the greatest part of the material becomes mobile and joins the liquid state. It is just natural to ask for the structural basis of the beta-relaxation: What units are moving around a which length scale and in which structural environment? This question has puzzled scientists for more than three decades. An answer to this question thus is of fundamental importance for this research field. Needless to say that this is of great significance for a huge number of processes that take place in liquid or glassy environments like a huge multitude of biological and chemical systems in nature.

The authors of the present manuscript have found a convincing answer to this question by employing the modern method of momentum-resolved time-domain interferometry to tackle this problem. This method, based on nuclear resonant analysis of synchrotron x-rays scattered from the sample, has the unique potential to reveal the dynamics of the sample on exactly the relevant time and length scales. The result is the finding that the elusive beta-relaxation near the glass transition is related to cage-breaking motions of the molecules of the material on the transition from the 'rattling-in-the-cage' process of fast relaxation to the structural alpha-relaxation. The method is convincingly explained, the technique is demonstrated experimentally and well documented. The manuscript constitutes an excellent piece of very original work to solve a problem that has been debated since decades. Therefore I consider this manuscript well suited for publication in Nature Communications.

Before publication, however, I would suggest a few improvements that enhance the accessibility of the material:

(1) In Fig. 4 it is not explained which of the curves belong to the alpha- and which belong to the beta-relaxation. This should be indicated directly in the graph.

(2) Since the microscopic picture of the Johari-Goldstein (beta-) relaxation is the central finding of the paper, the authors should introduce an extra figure that illustrates this finding, not in reciprocal

space, but in real space: How can the 'mosaic structure' that is even mentioned in the title of the manuscript, be visualized ?

Ralf Roehlsberger

Reviewer #3 (Remarks to the Author):

The manuscript reports an in-depth characterization, with both temporal and spatial resolution, of the dynamics of molecular glass former 1-propanol by gamma-resonance time-domain-interferometry (TDI) and dielectric spectroscopy (DS). Previously published data on other glass formers are considered too. The main outcome of the study is that the Johari-Goldstein (JG) beta relaxation is not just a local relaxation whose effects are spatially limited, but it is rather associated to a percolating cluster made by highly mobile connected molecules. Though several arguments are employed to attain this conclusion, the central one is based on the observation that the most probable mean-squared displacement of molecules participating to the JG beta relaxation – taken from the maximum of the imaginary part of the dielectric susceptibility – is large enough to fall in the range where structural instability takes place according to the Lindemann criterion.

The conclusions of the present study go beyond the traditional beliefs on the often neglected role in the glass transition of relaxations beyond the main alpha relaxation. The arguments provided to reach these conclusions have solid physical bases. The manuscript can be recommended for publication. However, before that the authors must address a number of concerns and modify the manuscript accordingly.

Reply

We thank the referee for the careful reading and summary of the manuscript and for recognising the physical basis of our conclusions. In the following her/his concerns and comments are addressed.

Questions/comments:

*1. The premise of the work is based on whether the (JG) beta relaxation can be considered a mere spectator or rather is a driver of the glass transition. The conclusion that the latter holds must be contextualized within our current knowledge showing that not only the alpha process determines the glass transition: *Sci. Adv* 6, 1454 (2020); and that the landscape is much rougher than is classically assumed considering only the alpha relaxation as a driver of the glass transition: *Nat. Comm.* 5, 3725 (2014).*

We thank the referee for the two suggested references that are indeed relevant for contextualizing the role of the Johari-Goldstein relaxation within the glass-transition, especially in view of a growing number of evidences that the structural relaxation is not the only actor in the vitrification process. Accordingly, a paragraph has been added in the introduction section of the manuscript.

2. TDI experiments are conducted in proximity of the alpha-beta splitting. Why none of the density correlation functions under any condition provide any evidence for both the two relaxations?

One drawback of the current implementation of TDI is that its dynamical range spans less than two decades in time. This limitation, together with the stretched nature of the α and β_{JG} relaxations, made it impossible to probe more than one process simultaneously.

Specifically, at temperatures below $T_{\alpha\beta}$, when the β_{JG} process separates from the α one, the characteristic time of the structural relaxation becomes very quickly too slow, and TDI is only sensitive to the faster of the two relaxation processes, that is to the β_{JG} one.

Very close to the decoupling temperature $T_{\alpha\beta}$, the α and β_{JG} relaxations are expected to coexist in the probed time-window, but the accuracy of our experimental data does not allow us to disentangle the two (having similar timescale) through the fitting procedure.

A comment along this line has been added in the main text.

3. A crossover from alpha to beta relaxations is claimed at 15 nm⁻¹. However, I see no unambiguous signature of this whatsoever. Here it is safe to avoid speculating on such crossover considering that later in the manuscript (Fig. 3) the alpha and beta process dominance in each temperature range becomes clear.

We actually agree with the referee. We have then rephrased the discussion on the crossover at 15nm⁻¹ accordingly.

4. At the beginning of page 3, I would rather say that the q dependence of tau demonstrates the alpha and beta relaxation nature of the processes observed at 131.4 and 122.5 K.

Following the referee's suggestion, a sentence has been added, clarifying that the q-dependence of the relaxation time is crucial to identify the nature of the relaxation processes observed above and below 131.4K.

5. The average intermolecular distance is obtained from the molecular volume. This connection must be discussed in details.

As intermolecular distance we actually meant the center-of-mass to center-of-mass distance. This is now clarified in the text.

The average inter-molecular distance was estimated for all samples as $v_{mol}^{1/3}$, where v_{mol} is the molecular volume calculated from the sample density and molar weight. This procedure was preferred to the often-used estimation based on q_{max} . In fact, q_{max} identifies in the reciprocal space the strongest interatomic correlation and does not always reflect the average inter-molecular distance (center-of-mass to center-of-mass).

For example, in OTP the main peak of the $S(q)$ ($q_{max} = 1.4 \text{ \AA}^{-1}$) arises from the correlations between phenyl rings belonging to different molecules while nearest-neighbours center-of-mass correlations are encoded in a pre-peak located at a lower q value (0.85 \AA^{-1}). Taking this into account, the estimation based on the molecular volume provides values of the inter-molecular distance compatible with the results provided by the partial pair distribution functions (which are available only for 1-propanol and OTP, though at higher temperatures than those investigated by us, but not for 5-methyl-2-hexanol), namely:

	$v_{mol}^{1/3}$	$g(r)$
OTP (300K)	7 Å	7.7 Å
1-propanol (300K)	5 Å	5.3 Å

6. All considerations regarding the mean-squared displacement of molecules taking part to the JG beta relaxation only account for the most probable relaxation time in DS. Why not considering the mean-squared displacement of the whole distribution of tau_betaJG? This should provide additional important information on the fraction of the beta JG relaxation fulfilling the Lindemann criterion for structural instability.

We definitively agree with the referee that it would be very interesting to map the distribution of characteristic times of the Johari-Goldstein relaxation onto the distribution of mean-squared displacements associated to all molecules participating in it. Unfortunately, in order to do that properly we need to know in detail at least the q -dependence of both the characteristic time of the process and of its relaxation strength. At the moment, we could only probe the characteristic time and the relaxation strength in a q -range spanning less than half-a-decade. So, while we find reasonable to conclude that around the most-probable relaxation time identified by the dielectric spectroscopy measurements the Gaussian approximation holds, we cannot argue the same in particular at: 1) longer times (where the fast components of the structural relaxation will start to influence the molecular dynamics) and 2) at longer length-scales (where the dynamics of the Johari-Goldstein relaxation is expected to become more complex and cooperative, perhaps involving string-like collective rearrangements). For this reason, we have limited the discussion on the mean-squared displacement around the length- and time-scale identified by the most-probable relaxation time.

Hopefully we will be able to address this interesting issue once we will manage to extend the TDI experiments to a larger q -range, but this will require a relevant experimental upgrade.

7. I found rather peculiar that the authors employ the Cole-Cole equation to fit the beta JG relaxation and then claim that the beta JG relaxation can be described by the KWW equation. It is well known that the two descriptions are not equivalent (rather the Havriliak-Negami equation, under certain conditions, is equivalent to the KWW equation). If the beta JG relaxation is symmetric, I see no way how the KWW could be valid.

The KWW expression, which is commonly used in TDI experiments on glass-formers (see Ref. [9–10,12] in the manuscript), was used in order to introduce a minimal amount of bias into the analysis of the experimental data across $T_{\alpha\beta}$ and in the determination of the crossover temperature. We also observe that the exact shape of the JG-relaxation process for what concerns density fluctuations in principle might be different from the one appearing in DS measurements.

Moreover, while of course we agree with the referee's comment on the clear differences between the Cole-Cole and KWW expressions, we would like to point out that even in case density fluctuations within the JG-relaxation were described by the Cole-Cole function with the same shape parameter as in dielectric spectroscopy measurements (i.e. $a=0.50$), the corresponding relaxation function (described in time by the Mittag-Leffler function ([FCAA 19(5), 1105 (2016)])) would not differ much from the KWW equation up to one characteristic time. For instance, if we consider the TDI beating pattern measured

Figure 1: Time-domain interferometry data as a function of time (squares with error-bars) at 122.5K and 25nm^{-1} . The fitting curves obtained modelling the relaxation function in terms of the KWW and Cole-Cole equations are reported as a black dashed-line and a blue solid line, respectively. The residuals of the corresponding fit procedures are plotted on the bottom panel with the same colors as in the main panel.

at $q = 25\text{nm}^{-1}$ and $T = 122.5\text{K}$, sensitive only to the Johari-Goldstein relaxation, and we fit this data-set using a Mittag-Leffler function (Cole-Cole in frequency) for the decay of the beating pattern contrast, we obtain analogous results as for the analysis based on the KWW equation ($\beta_{KWW} = 0.66$): the initial beating pattern contrast, f'_q , and the characteristic time, τ , are compatible within one standard deviation, see below.

	τ [ns]	f'_q
KWW	$(5.3 \pm 1.9) \cdot 10^2$	0.54 ± 0.02
Cole-Cole	$(6 \pm 3) \cdot 10^2$	0.60 ± 0.04

The fitting curves and the residuals are also very close, see Fig. 1, and the calculated contrast functions are, within the time window directly probed by TDI, again in agreement within one standard deviation, see Fig. 2. This is also consistent with our previous work on 5-methyl-2-hexanol (Ref. [12]

Figure 2: model contrast functions, $\phi'(t, q)$, calculated from the parameters of the fits shown in Fig. 1. The KWW and the Cole-Cole relaxation functions are reported in red and blue, respectively, together with their one standard deviation uncertainty. The vertical black dot-dashed lines indicate the time-window directly accessed by TDI.

in the manuscript) where we have shown that the KWW equation provides a good description of the relaxation process, within experimental accuracy, for the data collected in the T -range (165K – 181K) and q -range ($31\text{nm}^{-1} - 40\text{nm}^{-1}$) where the β_{JG} -process dominates the relaxation of density fluctuations. Fitting the curves using the Cole-Cole relaxation function is instead more cumbersome because: i) the Mittag-Leffler function has no simple analytical form; ii) the dynamic range of TDI is less than two decades, and therefore our TDI spectra are quite insensitive to the long-time tail of the Mittag-Leffler function, that is at times

where the KWW and the Mittag-Leffler functions are clearly different, see Fig.2.

Therefore, despite using a KWW function to describe a relaxation that might be in principle Cole-Cole in shape, we consider our results to be robust, at least at timescales up to the characteristic time of the Johari-Goldstein relaxation. A comment summarizing these considerations has been added in the Methods section where the fitting models are discussed.

8. Anyhow the all β_{KWW} values must be provided. If, as reported in the SI, $n \cdot \beta_{KWW} = 2$, then β_{KWW} should be in all cases approximately 0.5

Following the request of the referee, all the β_{KWW} values used in the analysis of the propanol data are explicitly indicated in the Methods. Moreover, a table, reporting the β_{KWW} used in the fits at $q=q_{DS}$ as well as the values obtained for n has been added to the S.I. (section S2) where the connection between q_{DS} and the mean squared fluctuation is discussed. Furthermore, a figure plotting $n \cdot \beta_{KWW}(q_{DS})$ for the three analysed samples has also been added (Fig. S4 in the S.I.), showing that in of all the considered cases this product is indeed approximately equal to 2.

Reviewer #5 (Remarks to the Author):

The manuscript deals with the solution of a problem in the physics of liquids and glasses, namely the microscopic basis for the so-called Johari-Goldstein relaxation, also known as beta-relaxation. This characterizes the dynamical behavior of a liquid upon the transition into the glassy state. This dynamic sets in significantly before a glass turns into a liquid when being warmed up, i.e., the beta relaxation precedes the so-called alpha-relaxation where the greatest part of the material becomes mobile and joins the liquid state. It is just natural to ask for the structural basis of the beta-relaxation: What units are moving around a which length scale and in which structural environment? This question has puzzled scientists for more than three decades. An answer to this question thus is of fundamental importance for this research field. Needless to say that this is of great significance for a huge number of processes that take place in liquid or glassy environments like a huge multitude of biological and chemical systems in nature.

The authors of the present manuscript have found a convincing answer to this question by employing the modern method of momentum-resolved time-domain interferometry to tackle this problem. This method, based on nuclear resonant analysis of synchrotron x-rays scattered from the sample, has the unique potential to reveal the dynamics of the sample on exactly the relevant time and length scales. The result is the finding that the elusive beta-relaxation near the glass transition is related to cage-breaking motions of the molecules of the material on the transition from the 'rattling-in-the-cage' process of fast relaxation to the structural alpha-relaxation. The method is convincingly explained, the technique is demonstrated experimentally and well documented. The manuscript constitutes an excellent piece of very original work to solve a problem that has been debated since decades. Therefore I consider this manuscript well suited for publication in Nature Communications.

Reply:

We thank the Reviewer for the careful reading and effective summary of our manuscript and for considering it appropriate for publication in Nature Communications.

In the following we address the concerns raised by the Reviewer.

Questions/comments:

(1) In Fig. 4 it is not explained which of the curves belong to the alpha- and which belong to the beta-relaxation. This should be indicated directly in the graph.

Arrows with the required information have now been added in order to improve the figure readability and help the reader to identify the two different processes.

(2) Since the microscopic picture of the Johari-Goldstein (beta-) relaxation is the central finding of the paper, the authors should introduce an extra figure that illustrates this finding, not in reciprocal space, but in real space: How can the 'mosaic structure' that is even mentioned in the title of the manuscript, be visualized ?

We thank the referee for this idea. Following her/his advice, a new figure has been added (Fig. 5) in the main text in order to help visualizing the "mosaic structure" discussed in the manuscript. Fig. 5 provides a sketch in real-space of the distribution of the molecules participating to the JG-relaxation at a given time. These molecules (plotted as dark red spheres) are highly mobile so that, after a time of the order τ_{JG} , perform spatial excursions of the order of those prescribed by the Lindemann

criterion for crystal instability and larger than the rest of the molecules (white spheres) and ii) are spatially connected in a close to percolating cluster as deduced from the Johari-Goldstein relaxation strength of 0.25 as discussed in the text

REVIEWER COMMENTS

Reviewer #3 (Remarks to the Author):

The rebuttal to almost all reviewers' comments are generally satisfactory and the manuscript has been considerably improved. Regarding the authors' response to comment 6 of my previous report, it is true that TDI allows exploring a q range of only half a decade. However, via equation 1, this gives a relaxation time span of about 2 decades. This should allow determining the mean-squared displacement of the β_{JG} process over a significant range around the maximum of the imaginary part of the dielectric susceptibility and not only to the τ (β_{JG}) corresponding to that maximum. In turns, it should be possible establishing a threshold of τ (β_{JG}) below which the Lindemann criterion for stability is fulfilled. Before publication of the manuscript, I would like to hear the authors' viewpoint, and possibly see the related modification to the manuscript, on this extension of their analysis to a range of τ (β_{JG}) larger than just that of the peak of the imaginary part of the dielectric susceptibility. If this extension is possible, I am convinced this would provide an even larger impact to this work.

Reviewer #5 (Remarks to the Author):

The authors have satisfactorily addressed my comments and, in my view, also those of the other reviewer. Therefore I can now recommend the manuscript for publication in Nature Communications. Especially the addition of Fig 5 enhances the readability and comprehensibility of the content.

Another minor thing still to correct: In Fig. 4 the labels (a) and (b) are missing in the figure.

Reviewer #3 (Remarks to the Author):

The rebuttal to almost all reviewers' comments are generally satisfactory and the manuscript has been considerably improved.

Reply

We thank reviewer 3 for her/his second evaluation and for recognising the improvements of the manuscript. In the following we address the last point raised by the referee.

Questions/comments:

Regarding the authors' response to comment 6 of my previous report, it is true that TDI allows exploring a q range of only half a decade. However, via equation 1, this gives a relaxation time span of about 2 decades. This should allow determining the mean-squared displacement of the β_{JG} process over a significant range around the maximum of the imaginary part of the dielectric susceptibility and not only to the τ (β_{JG}) corresponding to that maximum. In turns, it should be possible establishing a threshold of τ (β_{JG}) below which the Lindemann criterion for stability is fulfilled. Before publication of the manuscript, I would like to hear the authors' viewpoint, and possibly see the related modification to the manuscript, on this extension of their analysis to a range of τ (β_{JG}) larger than just that of the peak of the imaginary part of the dielectric susceptibility. If this extension is possible, I am convinced this would provide an even larger impact to this work.

We agree with the referee that it would be interesting to associate the distribution of characteristic times for the Johari-Goldstein relaxation to the distribution of mean-squared displacements for the molecules (or rather local regions) contributing to it. In our manuscript we limited this discussion only to the most probable values of both distributions. The extension suggested by the referee would allow for instance the determination of: 1) the fraction of the "more mobile molecules" participating to the β_{JG} -relaxation, once a mobility threshold is assigned, and 2) the threshold, in terms of relaxation time, for cage-breaking events. This extension, however, requires some critical assumptions as discussed hereafter.

Eq. 1 in our manuscript establishes a connection between mean-squared displacement and characteristic time, as the referee points out. It relies on the hypothesis that density fluctuations within the Johari-Goldstein relaxation can be treated in the incoherent approximation. As discussed in the S.I, this requires that the relaxation strength of the process does not have a strong q -dependence. While we know this to be the case around q_{DS} , where, in fact, it was possible to measure the relaxation strength of the β_{JG} -relaxation for one of our samples (5-methyl-2-hexanol), we do not have information for $q \gg q_{DS}$ and $q \ll q_{DS}$. At the moment, therefore, we cannot really justify Eq. 1 for scattering vectors other than q_{DS} .

Furthermore, for what concerns the identification of the "threshold relaxation time", it would be necessary to know the exact value of the mean-squared displacement for the Lindemann criterion of instability for each of the considered liquid samples. This information is not available for amorphous structures and, in particular, for the transient structures forming in the deeply supercooled state. For this reason, the mean-squared displacement estimated for our samples has been compared in the main text with the range of typical values expected for structural instability.

A possible way out is of course to assume the normalized Lindemann root mean-squared displacement for cage breaking events to be the same as for the corresponding crystal at melting, where normalized means rescaled to the CM-CM distance. This value would likely change from sample to sample. As reported in the literature, for instance, different crystalline lattices have Lindemann parameters varying by almost a factor two [A. Cho, J. Phys. F: Met. Phys. 12, 1069 (1982)]. For the same reason, it is reasonable to expect different Lindemann parameters between the crystal and the glass.

So, summarizing what discussed above, in order to exploit Eq. 1 in the manuscript to express the distribution of relaxation times corresponding to the JG process in terms of a distribution of mean-squared displacements and identify the threshold relaxation time for structural instability, it is necessary to assume: 1) Eq. 1 to be valid at all q -values, and 2) that the Lindemann mean-squared displacement at melting is exactly the same as for cage-breaking events in the deep supercooled state.

Under these two assumptions we can extend our analysis at least for the case of o-terphenyl, the only sample among the investigated ones for which the Lindemann limiting mean-squared displacement is known. More precisely, it can be computed extrapolating at the melting point elastic neutron scattering data available in the literature [Tölle A., et al. *Eur. Phys. J. B* **16**, 73 (2000)], see the figure below.

If the square root of this value is rescaled by the average CM-CM distance (see S.I.) a ratio of 0.113(2) is obtained, which is rather close to (and, actually, compatible within 1 SD with) our estimate for the most-probable CM displacement within the JG-relaxation for o-terphenyl. In fact, the mean of the values obtained at 244 K and 265 K (see Fig. 4 in the manuscript) is $\left\langle \frac{\Delta r_{JG}}{r_p} \right\rangle = 0.125(17)$. Therefore, $\tau_{\beta_{JG}^{DS}}$ seems to identify the exact timescale at which the critical MSD is reached for the molecules participating to the JG-relaxation. If we look at the JG-relaxation as a collection of elementary processes with different timescales, $\tau_{\beta_{JG}^{DS}}$ would then discriminate which processes satisfy the criterion for structural instability (those with $\tau \geq \tau_{\beta_{JG}^{DS}}$) and which don't (those with $\tau < \tau_{\beta_{JG}^{DS}}$).

In order to calculate the fraction of such structurally unstable, “more mobile” molecules, the knowledge of the mean-squared displacement distribution would be instead required. A possible way to derive it is to associate a distribution of MSD to the distribution of relaxation times deduced e.g. from dielectric spectroscopy. In order to do that we need however to make further critical assumptions, the most delicate one being to assume that there exists a one-to-one relation between relaxation time and mean-squared displacement.

More in detail, we could assume that the Johari-Goldstein relaxation consists of i) independent and ii) locally exponential processes. It would then be possible to write the dielectric susceptibility in terms of a distribution of relaxation times, $G(\log_{10}(\tau))$:

$$\chi(\omega) = \int_0^{+\infty} G(\log_{10}(\tau)) d \log_{10}(\tau) \frac{1}{1 - i\omega\tau}$$

$G(\ln\tau)$ has an analytical form for the Cole-Cole susceptibility (Ref. [4] in the manuscript) and therefore it is possible, exploiting Eq.1, to rewrite it in terms of Δr_{JG} , i.e. $G(\log_{10} \Delta r_{JG})$. The Figure below reports $G(\log_{10} \Delta r_{JG})$ calculated in this way for o-terphenyl at 265 K. The distribution is symmetric around $\Delta r_{JG}(\tau_{\beta_{JG}}^{DS}) = \frac{\sqrt{6}}{q_{DS}}$ which, following the discussion above, would represent the threshold for structural instability. Since, with the assumptions discussed above, $\Delta r_{JG}(\tau_{\beta_{JG}}^{DS})$ also corresponds to the median of the distribution of normalized root mean-squared displacements, we can then conclude that the fraction of molecules relaxing via the JG-relaxation and satisfying the Lindemann criterion for instability at $\tau_{\beta_{JG}}^{DS}$ is, for the case of OTP, equal to 50% (see the red dashed area). For other materials $\frac{\Delta r_{JG}}{r_p}$ might not match the Lindemann threshold for structural instability, and the estimate for this fraction might be different.

However, it is important to stress that this number strongly depends on the actual shape of the distribution, which we do not know enough in detail. In fact, the grey area in the Figure below highlights the limited range of MSD directly accessed in our experiment. It clearly appears not sufficient for evaluating the detailed shape of the distribution and validate the previous assumption. For instance, no reliable estimates for neither the skewness nor the tails of the distribution are available. Moreover, simulations on metallic glasses suggest a rather complex shape for the distribution of atomic displacements within the Johari-Goldstein relaxation (see Yu, H. B et al., Structural rearrangements governing Johari-Goldstein relaxations in metallic glasses. *Sci. Adv.* 3 1701577 (2017)).

In conclusion, in order to estimate a threshold value for the characteristic time $\tau_{\beta_{JG}}$ above which the Lindemann criterion for structural instability is fulfilled it is necessary to assume that that the exact value for the Lindemann criterion in the case of the supercooled liquid matches that for its crystalline counterpart. Further hypotheses are then required to calculate the fraction of molecules satisfying the criterion for structural

instability, and namely that: i) the incoherent approximation (and therefore Eq. 1 in the main text) is valid over the whole q -range; ii) the Johari-Goldstein relaxation process can be described in terms of independent, exponential relaxations; iii) a one-to-one correspondence holds between the relaxation times extracted from DS measurements and the MSD.

Since it is hard for us to justify all of these assumptions at this stage, we hesitate to include this whole discussion in our manuscript. However, we do believe that the reflection that the referee was urging us to make opens interesting scenarios, and therefore we have included in the supplementary materials accompanying this manuscript a paragraph which points out that, for the case of OTP: i) the Lindemann root mean-squared displacement at melting happens to match the root mean-squared displacement at $\tau_{\beta_{JG}}^{DS}$ of the molecules participating to the Johari-Goldstein process, and that therefore ii) all the molecules relaxing with a timescale equal or longer than $\tau_{\beta_{JG}}^{DS}$ will satisfy the Lindemann criterion for structural instability. We also clarify in the text that this conclusion remains to be demonstrated for other glass-formers.

Reviewer #5 (Remarks to the Author):

The authors have satisfactorily addressed my comments and, in my view, also those of the other reviewer. Therefore I can now recommend the manuscript for publication in Nature Communications. Especially the addition of Fig 5 enhances the readability and comprehensibility of the content.

Reply

We appreciate Reviewer #5 recognized our effort to improve the manuscript following her/his advice and we thank her/him for supporting the publication of our manuscript.

Questions/comments:

Another minor thing still to correct: In Fig. 4 the labels (a) and (b) are missing in the figure.

Labels (a) and (b) have now been added to Fig. 4.

REVIEWERS' COMMENTS

Reviewer #3 (Remarks to the Author):

The response provided by the authors on my last concern is satisfactory. The extension to a larger gap of time/length scale would require a level of detail and effort that goes beyond the scope of this study. I look forward to seeing further development of the very interesting insights contained in the manuscript. In the meantime, I recommend publication of the manuscript in its present form.

Reviewer #3 (Remarks to the Author):

The response provided by the authors on my last concern is satisfactory. The extension to a larger gap of time/length scale would require a level of detail and effort that goes beyond the scope of this study. I look forward to seeing further development of the very interesting insights contained in the manuscript. In the meantime, I recommend publication of the manuscript in its present form.

Reply

We appreciate that Reviewer #3 recognized the potential significance of our results and our effort to improve the manuscript following her/his advises and we thank her/him for supporting the publication of our manuscript.